# A simulation-based method to inform serosurvey design for estimating the force of infection using existing blood samples

Anna Vicco[1,2], Clare P. McCormack[2], Belen Pedrique[3], John H. Amuasi[4,5,6,7], Anthony Afum-Adjei Awuah[5,6,8], Christian Obirikorang[5,6,8], Nicole S. Struck[9,10], Eva Lorenz[9,10,11], Jürgen May[9,10,12], Isabela Ribeiro[3], Gathsaurie Neelika Malavige[3], Christl A. Donnelly[2,13,14], Ilaria Dorigatti[2]*

1 Department of Molecular Medicine, University of Padua, Padua, Italy, 2 MRC Centre for Global Infectious Disease Analysis, School of Public Health, Imperial College London, London, United Kingdom, 3 Drugs for Neglected Diseases initiative, Geneva, Switzerland, 4 Department of Global Health, School of Public Health, Kwame Nkrumah University of Science and Technology, Kumasi, Ghana, 5 Global Health and Infectious Diseases Research Group, Kumasi Centre for Collaborative Research in Tropical Medicine, Kumasi, Ghana, 6 Research Group Global One Health, Department of Implementation Research, Bernhard Nocht Institute of Tropical Medicine, Hamburg, Germany, 7 Division for Tropical Medicine, Department of Medicine, University Medical Centre Hamburg-Eppendorf, Hamburg, Germany, 8 Department of Molecular Medicine, School of Medicine and Dentistry, Kwame Nkrumah University of Science and Technology, Kumasi, Ghana, 9 Infectious Disease Epidemiology, Bernhard Nocht Institute for Tropical Medicine, Hamburg, Germany, 10 German Center for Infection Research (DZIF), partner site Hamburg-Borstel-Lübeck-Riems, Germany, 11 Institute of Medical Biostatistics, Epidemiology and Informatics, University Medical Centre of the Johannes Gutenberg University Mainz, Mainz, Germany, 12 Department of Tropical Medicine I, University Medical Center Hamburg-Eppendorf (UKE), Hamburg, Germany, 13 Department of Statistics, University of Oxford, Oxford, United Kingdom, 14 Pandemic Sciences Institute, University of Oxford, Oxford, United Kingdom

* i.dorigatti@imperial.ac.uk

**Data Availability Statement:** All data and code are available in the Supporting information files.

## Abstract

The extent to which dengue virus has been circulating globally and especially in Africa is largely unknown. Testing available blood samples from previous cross-sectional serological surveys offers a convenient strategy to investigate past dengue infections, as such serosurveys provide the ideal data to reconstruct the age-dependent immunity profile of the population and to estimate the average per-capita annual risk of infection: the force of infection (FOI), which is a fundamental measure of transmission intensity.

In this study, we present a novel methodological approach to inform the size and age distribution of blood samples to test when samples are acquired from previous surveys. The method was used to inform SERODEN, a dengue seroprevalence survey which is currently being conducted in Ghana among other countries utilizing samples previously collected for a SARS-CoV-2 serosurvey.

The method described in this paper can be employed to determine sample sizes and testing strategies for different diseases and transmission settings.

**Funding:** CM, CAD and ID acknowledge funding from the Drugs for Neglected Diseases initiative and the MRC Centre for Global Infectious Disease Analysis (reference MR/X020258/1), funded by the UK Medical Research Council (MRC). This UK funded award is carried out in the frame of the Global Health EDCTP3 Joint Undertaking. AV thanks the Foundation Blanceflor Boncompagni Ludovisi for funding her PhD visiting period at Imperial College London. ID acknowledges research funding from a Sir Henry Dale Fellowship funded by Wellcome Trust (grant 213494/Z/18/Z). CAD thanks the UK National Institute for Health and Care Research Health Protection Research Unit (NIHR HPRU) in Emerging and Zoonotic Infections in partnership with Public Health England (PHE) for funding (grant HPRU200907). DNDi thanks the French Development Agency (AFD), France; Médecins Sans Frontières International; Swiss Agency for Development and Cooperation (SDC), Switzerland; UK aid, UK; for the financial support in this work. The funders had no role in study design, data collection and analysis, decision to publish, or preparation of the manuscript. The findings and conclusions contained herein are those of the authors and do not necessarily reflect positions or policies of the aforementioned funding bodies.

**Competing interests:** The authors have declared that no competing interests exist.

## Author summary

The historical circulation of dengue virus is still poorly understood in many parts of the world, and age-stratified seroprevalence surveys can provide the data to quantify population exposure to dengue and its transmission intensity.

In this work, we developed a simulation-based method that can be used to identify the sample sizes and age-distribution of the samples needed to obtain informative estimates of dengue force of infection from existing blood samples. We apply this method to data obtained from a SARS-CoV-2 serological survey previously conducted in three cities in Ghana.

The methods and code developed in this paper can be used to design serological surveys for dengue and other pathogens when using existing blood samples with accompanying information on age and location.

## Introduction

Dengue virus (DENV) is one of the most rapidly spreading vector-borne diseases worldwide and places a large burden on public health [1]. In the last 50 years, dengue incidence has increased more than 30-fold [2], with recent estimates suggesting that up to 105 million people are infected each year [3]. Due to ongoing changes in climate, demography and socioeconomic structure and growing urbanization [4], there are concerns that the geographical range of dengue will expand in the coming decades.

Estimates suggest this potential geographic expansion may affect Sub-Saharan Africa in particular [5–7], a region where the burden of dengue remains poorly understood and is highly uncertain [8,9]. There are also key knowledge gaps on the rates of asymptomatic infection [4,10] and misclassification [2]. In addition, the lack of a strong surveillance system and insufficient infrastructure, have been identified as key challenges for arbovirus surveillance and disease control across the African region [11].

Cross-sectional serological surveys, which involve the collection of sera samples and the subsequent analysis of antibody level in the sera [12], are the gold standard for assessing individual-level exposure and for understanding the historical circulation of dengue at the population-level [13]. Age-stratified serological surveys provide the necessary data to reconstruct the immunity profile of the population, i.e., the extent of dengue exposure as a function of age, which in turn can be used to estimate the force of infection (FOI), defined as the per capita rate at which susceptible individuals become infected [10,14], the burden of infection and the impact of interventions [3].

However, several challenges, including the need of substantial resources and infrastructure, have posed limitations on the number of serosurveys conducted in low-income countries [12]. In Africa for example, only 17 age-stratified dengue seroprevalence surveys have been conducted to date [3,15]. Optimizing the serosurvey design, particularly the number of samples to be tested, is critical to promoting the use of these surveys, especially when resources are limited. While conducting new seroprevalence surveys provides the opportunity to target optimal age-groups for estimating epidemiological parameters, it also requires extensive resources, including time and operational capacity. In some cases these requirements can be mitigated by leveraging existing blood samples.

Simulation-based analyses are a powerful tool for informing serosurvey design as they enable the testing of different sampling scenarios, which in turn can help determine sample

sizes targeted to the outcomes of interest [16–18]. For example, reversible catalytic models have been used to optimize the sample sizes required to accurately estimate the seroconversion rate to malaria [16], compartmental dynamic transmission models have been developed to identify the optimal age distribution of blood samples for different pathogens and outcomes of interest [17] and during the COVID-19 pandemic, Bayesian methods were developed to propagate the uncertainty of imperfect testing and determine optimal sample size to minimize the uncertainty in the seroprevalence estimates [18].

In this study, we developed methods to inform serosurvey design when leveraging existing blood samples. We use simple catalytic models to develop a new simulation-based framework for identifying the optimal number and age distribution of available blood samples to obtain accurate setting-specific dengue FOI estimates whist optimising resources and at the same time accounting for imperfect test sensitivity and specificity. The methods described in this study have been used to inform the design of serosurveys aimed to reconstruct dengue FOI estimates in three locations in Ghana [19], utilising blood samples previously collected for SARS-CoV-2 surveillance. Below we provide evidence of how the simulation framework developed in this study can be readily adopted for estimating the FOI of other infectious pathogens across different transmission settings.

## Materials and methods

### Ethics statement

Ethical approval was obtained from the Committee on Human Research and Publication (CHRPE) of the Kwame Nkrumah University of Science and Technology (KNUST), IRB number CHRPE/AP/218/20. Blood samples were collected from participants or legal guardians who gave informed written consent to participate in the study.

### Blood samples

The blood samples available for testing were collected in a SARS-CoV-2 seroprevalence study conducted in three cities in Ghana (Accra, Kumasi and Tamale) between February 2021 and February 2022 by the Bernard Nocht Institute for Tropical Medicine (BNITM) and the Kumasi Centre for Collaborative Research in Tropical Medicine (KCCR) [20]. The study was designed with a two-stage geographical cluster sampling method stratified by age and sex in urban households [20]. The available blood samples included 2,051 samples from participants aged 10 to 94 years; all samples were stored in a biobank at -80˚C.

### Potential sampling scenarios

We simulated alternative sampling scenarios (Fig 1) to select the sample sizes and their distribution. We denoted the scenario that included testing all available samples as 'scenario 0' and explored the following alternative scenarios: testing an equal number of samples in each age-group, using the minimum number of available samples by age-group (scenario A); testing all available samples in the younger age-groups (i.e., the first half of the age-groups), and half of the minimum number of samples across the young age-groups for the older age-groups (scenario B); testing all available samples for the older age-groups and half of the minimum number of samples across the old age-groups for the younger age-groups (scenario C); testing all available samples for the younger age-groups and half of the available number of samples for the older age-groups (scenario D) and testing all available samples for the older age-groups and half of the available number of samples for the younger age-groups (scenario E).

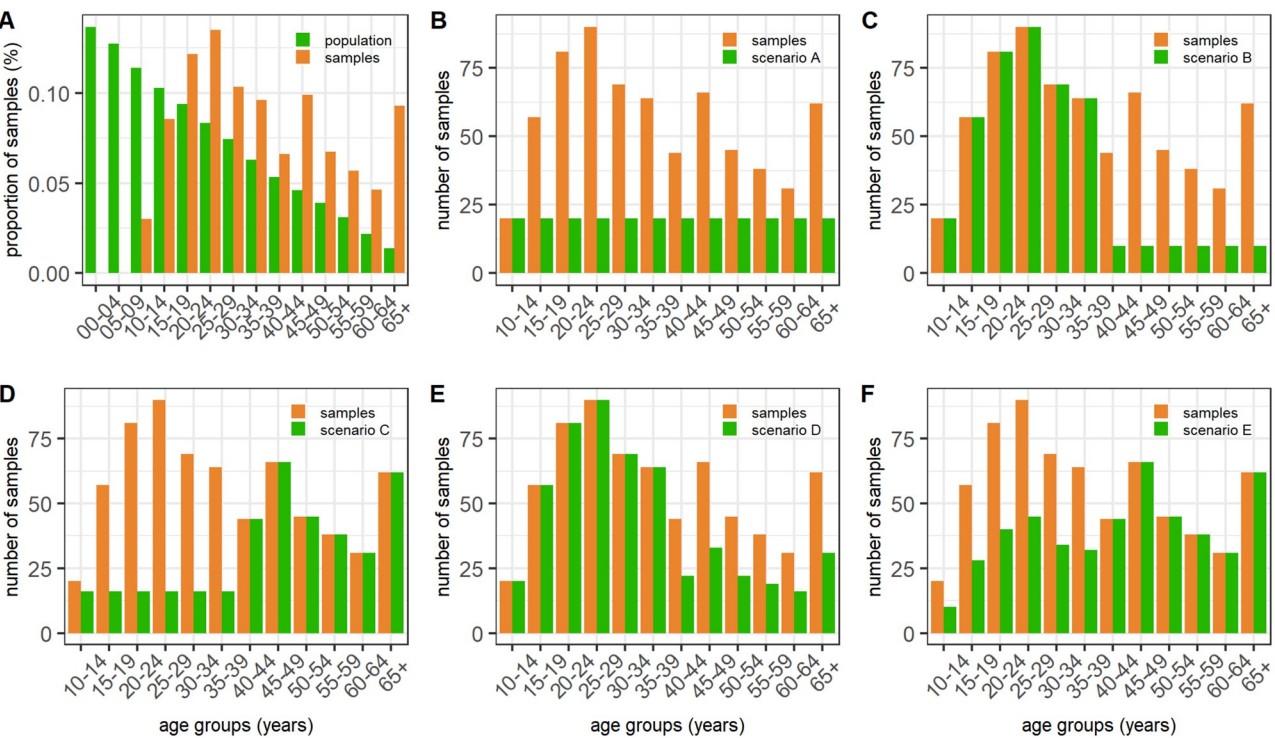

**Fig 1. Blood sample distribution across the simulated scenarios the for city of Accra using 5-year age groups.** Panel A shows the distribution of the available samples across the age-groups and compares them with the population age structure of Ghana (World Population Prospect [21]). Panels B to F illustrate the sample distribution under scenarios A to E compared to the available samples in the baseline scenario 0 (which in the panel is indicated as "samples").

## Simulation study & model fitting

In model 1, we assumed a constant in time and age average yearly FOI, and that antibodies do not wane. Using these assumptions, the age-dependent seroprevalence (i.e., the probability of developing measurable antibodies upon exposure) could be modelled as detailed in Eq 1, where $a_i$ is the midpoint of age-group $i$, $\lambda$ corresponds to the average yearly FOI per serotype, based on the estimates obtained from Cattarino et al. [3] (Table 1), and $n$ is the number of serotypes assumed to be circulating. For Ghana, we assumed $n = 2$ independently transmitting serotypes, following Bonney et al. [22].

$$z(a_i) = 1 - e^{-n\lambda a_i} \tag{1}$$

We assumed that the number of subjects exposed to dengue in each age-group was binomially distributed as a function of the number of samples tested in age-group $i$, as described in Eq 2

$$X_i \sim \text{Binomial}\left(N_i, z(a_i)\right) \tag{2}$$

where $X_i$ denotes the number of people testing seropositive in group $i$, $N_i$ denotes the number of samples tested in age-group $i$, and $z(a_i)$ is the probability of testing seropositive in that age-group.

For each sampling scenario we denoted the age-specific sample sizes $N_1, \ldots, N_m$ in age-groups $1, \ldots, m$, and simulated the expected age-stratified seroprevalence as shown in Eq 1. For

**Table 1. Summary of accuracy metrics obtained for Ghana with 5-year age groups under the optimal scenarios across models and assumptions.**

| Model | Type | Parameter values | Median age of first infection | City | Bias (95% CrI) (in percent, %) | Coverage (95% CI) (in percent, %) | Uncertainty (95% CrI) | Scenario | N samples |
|---|---|---|---|---|---|---|---|---|---|
| 1 | Cattarino's FOI | $\lambda = 0.017$ | 29 | Accra | 5.2 (0.3, 17.2) | 95.2 (93.7, 96.3) | 0.005 (0.004, 0.006) | B | 441 |
| 1 | Cattarino's FOI | $\lambda = 0.020$ | 25 | Tamale | 4.3 (0.2, 14.8) | 95.8 (94.4, 96.9) | 0.005 (0.005, 0.006) | B | 524 |
| 1 | Cattarino's FOI | $\lambda = 0.019$ | 26 | Kumasi | 4.7 (0.2, 16.4) | 94.4 (92.8, 95.7) | 0.005 (0.004, 0.006) | D | 522 |
| 1 | High FOI | $\lambda = 0.034$ | 15 | Accra | 5.3 (0.2, 21.2) | 95 (93.5, 96.2) | 0.012 (0.009, 0.016) | B | 441 |
| 1 | High FOI | $\lambda = 0.040$ | 13 | Tamale | 5.7 (0.3, 21.9) | 94.3 (92.7, 95.6) | 0.014 (0.01, 0.02) | B | 524 |
| 1 | High FOI | $\lambda = 0.038$ | 13 | Kumasi | 5.8 (0.4, 20.3) | 96.3 (94.9, 97.3) | 0.014 (0.011, 0.019) | B | 431 |
| 1 | Low FOI | $\lambda = 0.009$ | 59 | Accra | 5.1 (0.2, 17) | 94.6 (93, 95.8) | 0.003 (0.002, 0.003) | E | 475 |
| 1 | Low FOI | $\lambda = 0.010$ | 50 | Tamale | 5.4 (0.3, 16.7) | 94.7 (93.1, 95.9) | 0.003 (0.003, 0.003) | E | 464 |
| 1 | Low FOI | $\lambda = 0.010$ | 53 | Kumasi | 4.4 (0.3, 14.4) | 93.9 (92.2, 95.2) | 0.002 (0.002, 0.003) | 0 | 678 |
| 2 | Cattarino's FOI, baseline decay | $\lambda = 0.017$ $\alpha = 0.02$ | 29 | Accra | 7.8 (0.3, 28.3) | 99.5 (98.8, 99.8) | 0.012 (0.008, 0.02) | B | 441 |
| 2 | Cattarino's FOI, baseline decay | $\lambda = 0.020$ $\alpha = 0.02$ | 25 | Tamale | 9.8 (0.6, 30) | 98.7 (97.8, 99.2) | 0.015 (0.009, 0.025) | E | 464 |
| 2 | Cattarino's FOI, baseline decay | $\lambda = 0.019$ $\alpha = 0.02$ | 26 | Kumasi | 10.5 (0.4, 32.5) | 99.5 (98.8, 99.8) | 0.016 (0.01, 0.026) | A | 264 |
| 2 | High FOI, baseline decay | $\lambda = 0.034$ $\alpha = 0.02$ | 15 | Accra | 11.1 (0.5, 36) | 98.7 (97.8, 99.2) | 0.031 (0.019, 0.052) | A | 240 |
| 2 | High FOI, baseline decay | $\lambda = 0.040$ $\alpha = 0.02$ | 13 | Tamale | 9.2 (0.4, 27.3) | 99.4 (98.7, 99.7) | 0.03 (0.018, 0.051) | B | 524 |
| 2 | High FOI, baseline decay | $\lambda = 0.038$ $\alpha = 0.02$ | 13 | Kumasi | 9.2 (0.4, 27.8) | 98.8 (97.9, 99.3) | 0.026 (0.016, 0.045) | B | 431 |
| 2 | Cattarino's FOI, high decay | $\lambda = 0.017$ $\alpha = 0.06$ | 29 | Accra | 13.1 (0.9, 46.8) | 99.9 (99.4, 100) | 0.025 (0.015, 0.037) | A | 240 |
| 2 | Cattarino's FOI, high decay | $\lambda = 0.020$ $\alpha = 0.06$ | 25 | Tamale | 14.4 (0.5, 40.8) | 100 (99.6, 100) | 0.031 (0.021, 0.043) | C | 260 |
| 2 | Cattarino's FOI, high decay | $\lambda = 0.019$ $\alpha = 0.06$ | 26 | Kumasi | 13.6 (0.8, 42.9) | 99.5 (98.8, 99.8) | 0.025 (0.015, 0.038) | A | 264 |
| 3 | Decreasing FOI | $\lambda_1 = 0.013$ $\lambda_2 = 0.009$ $\lambda_3 = 0.004$ | 24 | Accra | 23 (1.3, 171.4) | 97.6 (96.5, 98.4) | 0.013 (0.008, 0.023) | C | 382 |
| 3 | Decreasing FOI | $\lambda_1 = 0.015$ $\lambda_2 = 0.010$ $\lambda_3 = 0.005$ | 21 | Tamale | 27.7 (0.9, 253) | 97.8 (96.7, 98.5) | 0.018 (0.011, 0.046) | C | 260 |
| 3 | Decreasing FOI | $\lambda_1 = 0.014$ $\lambda_2 = 0.009$ $\lambda_3 = 0.005$ | 22 | Kumasi | 23.8 (1, 172) | 97.7 (96.6, 98.5) | 0.014 (0.008, 0.024) | C | 415 |
| 3 | Increasing FOI | $\lambda_1 = 0.004$ $\lambda_2 = 0.009$ $\lambda_3 = 0.013$ | 38 | Accra | 22.3 (1, 72.6) | 95.4 (93.9, 96.5) | 0.011 (0.004, 0.022) | C | 382 |
| 3 | Increasing FOI | $\lambda_1 = 0.005$ $\lambda_2 = 0.010$ $\lambda_3 = 0.015$ | 32 | Tamale | 27.5 (1.7, 89.4) | 96.6 (95.3, 97.6) | 0.015 (0.006, 0.039) | C | 260 |
| 3 | Increasing FOI | $\lambda_1 = 0.005$ $\lambda_2 = 0.009$ $\lambda_3 = 0.014$ | 35 | Kumasi | 19.2 (0.9, 68.2) | 95.5 (94, 96.6) | 0.012 (0.004, 0.023) | C | 415 |

*(Continued)*

**Table 1.** (Continued)

| Model | Type | Parameter values | Median age of first infection | City | Bias (95% CrI) (in percent, %) | Coverage (95% CI) (in percent, %) | Uncertainty (95% CrI) | Scenario | N samples |
|---|---|---|---|---|---|---|---|---|---|
| 3 | *Non-monotonic FOI* | $\lambda_1 = 0.013$ $\lambda_2 = 0.004$ $\lambda_3 = 0.009$ | 29 | Accra | 18.6 (0.7, 119.4) | 98 (96.9, 98.7) | 0.011 (0.005, 0.023) | E | 475 |
| 3 | *Non-monotonic FOI* | $\lambda_1 = 0.015$ $\lambda_2 = 0.005$ $\lambda_3 = 0.010$ | 25 | Tamale | 21.8 (0.7, 165.4) | 95.9 (94.5, 97) | 0.013 (0.005, 0.048) | E | 464 |
| 3 | *Non-monotonic FOI* | $\lambda_1 = 0.014$ $\lambda_2 = 0.005$ $\lambda_3 = 0.009$ | 26 | Kumasi | 20.22 (0.9, 129.8) | 96.5 (95.2, 97.5) | 0.011 (0.006, 0.026) | E | 495 |

*In the High FOI scenarios we assumed an FOI twice as high as the Cattarino et al FOI, while in the Low FOI scenarios we assumed half the Cattarino et al FOI. The median age of first infection was calculated as $(1 / (n\lambda))$ where n is the number of circulating dengue serotypes, and in the time-varying scenario we estimated the average $\lambda$ as the population weighted average of $\lambda_1, \lambda_2, \lambda_3$ using the Ghanian age-structure reported in the UNWPP data [21].*

each scenario, we generated 1,000 datasets by sampling the number of seropositive samples $I_1, \ldots, I_m$ from a binomial distribution with parameters $N_1, \ldots, N_m$ and $z_1, \ldots, z_m$, where $z$ derives from Eq 1, assuming the FOI estimates from Cattarino et al. [3] (Table 1). To account for sensitivity (*se*) and specificity (*sp*) of the test respectively, we simulated the number of samples testing positive $P_1, \ldots, P_m$ among the $I_1, \ldots, I_m$ samples as the sum of the true positive $TP_1, \ldots TP_m$ and false positive $FP_1, \ldots, FP_m$ samples, which were sampled from a binomial distribution as described in Eqs 3–5.

$$P_i = TP_i + FP_i \tag{3}$$

$$TP_i \sim \text{Binomial}\ (I_i,\ se) \tag{4}$$

$$FP_i \sim Binomial\ (N_i - I_i, 1 - sp) \tag{5}$$

We assumed $se = 0.892$ and $sp = 0.988$ [23] and for each simulated dataset we fitted the catalytic model separately, thus obtaining 1,000 FOI estimates, denoted $\tilde{\lambda}$.

We account for imperfect testing by defining the test positivity function ($t^+(a_i)$) shown in Eq 6, which identifies the probability that an individual of age $a_i$ tests positive according to the seroprevalence, sensitivity (*se*) and specificity (*sp*) of the assay. We used a binomial log-likelihood (Eq 7) to infer the posterior distribution of the FOI $\lambda$ using the Hamiltonian Monte Carlo (HMC) algorithm coded in *CmdStanR [24]*, having assumed a uniform prior distribution between zero and one (Fig A in S1 Text), and sampling 3 chains of 5,000 iterations each, with a warming-up step of 1,000 iterations. Convergence was assessed visually and using the Rhat and effective sample size [24]. In the results we report the median and 95% credible interval of the posterior distribution of the FOI.

$$t^+(a_i) = se\ z(a_i)\ + (1 - sp)\ (1 - z(a_i)) \tag{6}$$

$$LnL(P_i\ |N_i, t^+(a_i)) = (P_i\ log(t^+(a_i))\ + (N_i - P_i)\ (log(1\ -\ t^+(a_i)) \tag{7}$$

## Performance metrics and selection criteria

Three performance indexes were used to compare the FOI estimates $\tilde{\lambda}$ with the FOI used to generate the simulated data. Specifically, we calculated the median of the FOI under each scenario and age category, and then calculated the:

- **Bias**: the absolute value of the difference between the estimated FOI $\tilde{\lambda}$ and the $\lambda$ used to simulate the data divided by $\lambda$, expressed in percentage.

- **Uncertainty**: the width of the 95% credible interval (CrI) of the estimated FOI $\tilde{\lambda}$.

- **Coverage**: the number of times the FOI $\lambda$ used to simulate the data falls within 95% CrI of the estimated FOI $\tilde{\lambda}$, across the 1,000 simulations, expressed in percentage.

We selected the optimal sample sizes and age-distributions based on the following criteria. Firstly, we retained only the scenarios with an upper bound of the 95% CrI of the bias within a 15% tolerance above the one obtained with scenario 0. Then, among these scenarios, we retained those with a coverage within 5% of the maximum coverage value across the selected scenarios. Finally, we selected the scenario with the smallest number of samples. We also compared *a posteriori* the uncertainty of the optimal scenario with the one of scenario 0 to check that the selected scenario did not introduce further uncertainty around the estimate.

## Sensitivity analyses

**Alternative categorisation used to report the age-specific seroprevalence.** In a sensitivity analysis, we ran the simulation study using 10-year age categories (Fig B in S1 Text) (instead of 5-year age categories as done in the main analysis) to explore the effect that reporting the age-specific seroprevalence over wider age-groups has on the precision of the FOI estimates.

**Alternative serocatalytic model formulations.** To adapt our framework to different pathogens and settings, we implemented the simulation framework for the case where antibody protection decays over time at a yearly rate α (model 2) [10]. In this scenario, the seroprevalence can be modelled as shown in Eq 8.

$$z(a_i) = \frac{\lambda}{\lambda + \alpha}\left[1 - e^{-n a_i(\lambda + \alpha)}\right] \tag{8}$$

We also explored scenarios with time-varying FOIs (model 3) [25]. In our simulation study we assumed that the FOI changed in a stepwise fashion over time (Table J in S1 Text), with individuals 0–29 years old (age-groups $i < 5$) exposed solely to $\lambda_1$, individuals 30–49 years old (age-groups $5 \leq i \leq 8$) exposed to $\lambda_1$ during the last 30 years of life and to $\lambda_2$ during the first 20 years of life and individuals 50+ years of age (age-groups $i > 8$) exposed to $\lambda_1$ during the last 30 years of life, $\lambda_2$ during the middle 20 years of life and $\lambda_3$ for the earlier years of their life. In this case the age-stratified seroprevalence can be described as shown in Eq 9:

$$z(a_i) = \begin{cases} 1 - e^{-n\lambda_1 a_i}, & i < 5 \\ 1 - e^{-n[\lambda_1 a_4 + \lambda_2(a_i - a_4)]}, & 5 \leq i \leq 8 \\ 1 - e^{-n[\lambda_1 a_4 + \lambda_2(a_8 - a_4) + \lambda_3(a_i - a_8)]}, & i > 8 \end{cases} \tag{9}$$

where $a_i$ is the midpoint of age-group $i$, $a_4$ is the mid-point of age-group $i = 4$ (25–29 years old), and $a_8$ is the mid-point of age-group $i = 8$ (45–49 years old).

We used models 2 and 3 to generate simulated datasets, and then used the corresponding catalytic model (respectively model 2 and 3) and the likelihood defined in Eq 7 to reconstruct

the posterior distribution of the FOI $\lambda$ and decay rate $\alpha$ in model 2 and of the three FOIs $\lambda_1$, $\lambda_2$ and $\lambda_3$ in model 3. We assumed a normal distribution with mean 0 and standard deviation of 1 for the lambda parameters in both model 2 and 3, and a normal distribution with mean 0.02 (as suggested in the literature [10]) and standard deviation of 0.5 for the decay rate in model 2. We sampled the parameters on the log scale to improve the exploration of the parameter space.

**Alternative assumptions on the magnitude of the FOI and decay rate.**　In model 1, we tested three different FOI values: 1) $\lambda$ taken from Cattarino et al. [3]; 2) twice the value taken from Cattarino et al. [3]; and 3) half the value taken from Cattarino et al. [3], as shown in Fig C in S1 Text.

In model 2, we applied our framework to three alternative scenarios exploring different FOI and decay rate values: scenario 1) $\lambda$ taken from Cattarino et al. [3] and a decay rate $\alpha$ = 0.02 from Imai et al. [10]; scenario 2) $\lambda$ equal to twice the estimate in Cattarino et al. [3] and $\alpha$ = 0.02; and scenario 3) $\lambda$ taken from Cattarino et al. [3] and a higher decay rate $\alpha$ = 0.06. Table 1 and Fig D in S1 Text show the values of $\lambda$ and $\alpha$ used in the three scenarios.

In model 3 we tested three different patterns of transmission, with increasing, decreasing, and non-monotonic FOIs as shown in Fig E in S1 Text. In the selection of the optimal sampling scenario, we calculated the average bias, coverage and uncertainty over the three FOIs.

## Results

A total of 2,051 blood samples from a previous seroprevalence study conducted in three cities in Ghana (Accra, Kumasi and Tamale) [20], were available for testing for dengue antibodies (Table 1, scenario 0). Younger age-groups were less represented in the study population than in the general population, with 24.6% of the participants belonging to the 10–24 years age-group, compared to an expected 41.2% from the national age structure published in the 2022 Revision of World Population Prospects (WPP) (Fig 1). The 20–34 years age-group was the most represented (39.3% of participants) [20].

Fig 2 shows the bias, uncertainty, and coverage obtained across all scenarios using the 5-year age categories for the three Ghanian cities under model 1 and the Cattarino et al. FOI estimates [3]. The corresponding figure for the 10-year age categories is presented in the Supporting Information (Fig B in S1 Text). Overall, we found that reducing the sample sizes did not bias the FOI estimates, except for scenario A (which had the smallest number of samples) where the central estimate was consistently larger. On the other hand, as expected, we found that reducing the sample sizes increased the uncertainty of the FOI estimates, depending on the number and distribution of the samples by age. On average, we observed a coverage between 93% and 96% across all analysed scenarios, with some variability depending on the city. Fig 2 shows that reducing the number of tests in certain age-groups (such as in scenario B, highlighted in green) can increase the coverage without biasing the estimates.

In Ghana, using the FOI estimates from Cattarino et al. [3] and the simple catalytic model (model 1), we found that scenarios B and D, which included fewer samples from the older age-groups, showed a higher coverage and a lower uncertainty compared to the baseline scenario (scenario 0). This indicates that selecting the appropriate age distribution of the samples can help improve the accuracy of the FOI estimates, while at the same time reducing the sample sizes. Scenarios B and D provided the best combination of coverage, bias and uncertainty in both the 5-year (Fig 2) and 10-year (Fig B in S1 Text) age categories for the three cities. In these scenarios, the distribution of samples in the two age categories led to a similar uncertainty, but the coverage was generally higher in the 5-year age category, which also required testing fewer samples than with the 10-year group (a total of 1,487 instead of 1,610). As a result,

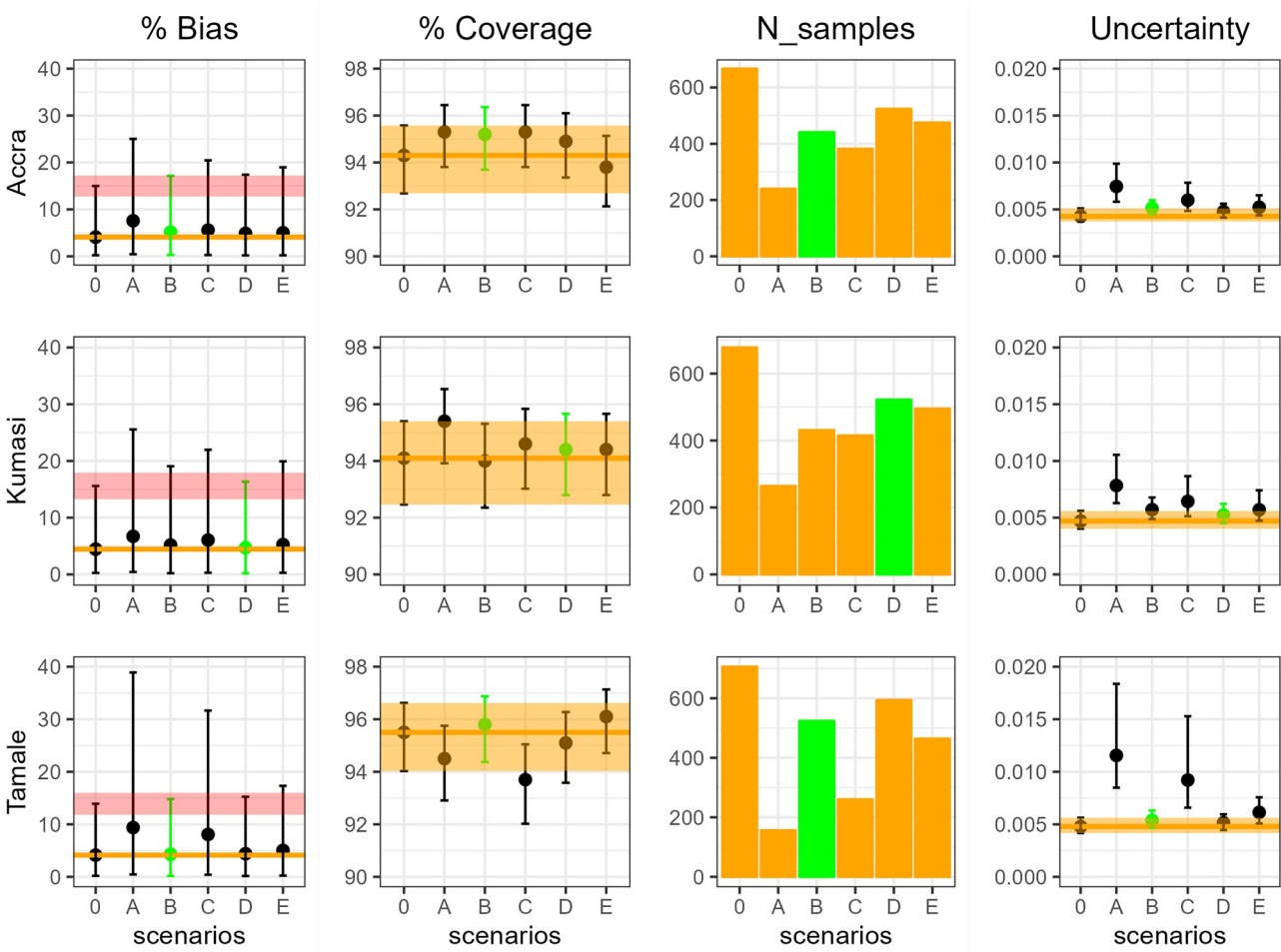

**Fig 2. Summary of the accuracy metrics obtained for Accra, Kumasi and Tamale with 5-year age category across scenarios.** The four columns represent respectively the bias (in percentage), coverage (in percentage), number of tested samples and uncertainty obtained for each scenario. The scenario highlighted in green indicates the selected scenario. The median bias and uncertainty are reported with their 95% CrI (columns 1 and 4), while the median coverage is reported with its 95% exact binomial CI. The orange line represents the median (columns 1, 2 and 4) and the orange ribbon represents the 95% CrI of the baseline scenario 0 (columns 2 and 4). The pink ribbon in the first column represents the 15% tolerance around the upper bound of the 95% CrI of scenario 0, which was used in the first step of the selection criterion.

using the proposed selection criterion, the total sample size of the study in Ghana was reduced to 1,487 samples from the available 2,051 samples.

Fig 3 shows the expected seroprevalence profile and model fit obtained from the baseline scenario (scenario 0) versus those obtained from the selected scenario, indicating no significant deviation in model fit caused by the reduced samples sizes and adopted sampling strategy.

## The optimal testing scenario changes with the expected FOI

In a sensitivity analysis we tested the impact of assuming alternative FOI values for the three locations, as described in Table 1. Fig 4 summarises the results obtained under a high and low FOI, showing that in higher FOI settings testing younger age-groups is preferrable (scenario B), while in lower FOI settings testing of older age-groups (scenario E) is required to get accurate FOI estimates. In both cases the chosen scenario allowed to estimate the FOI with high

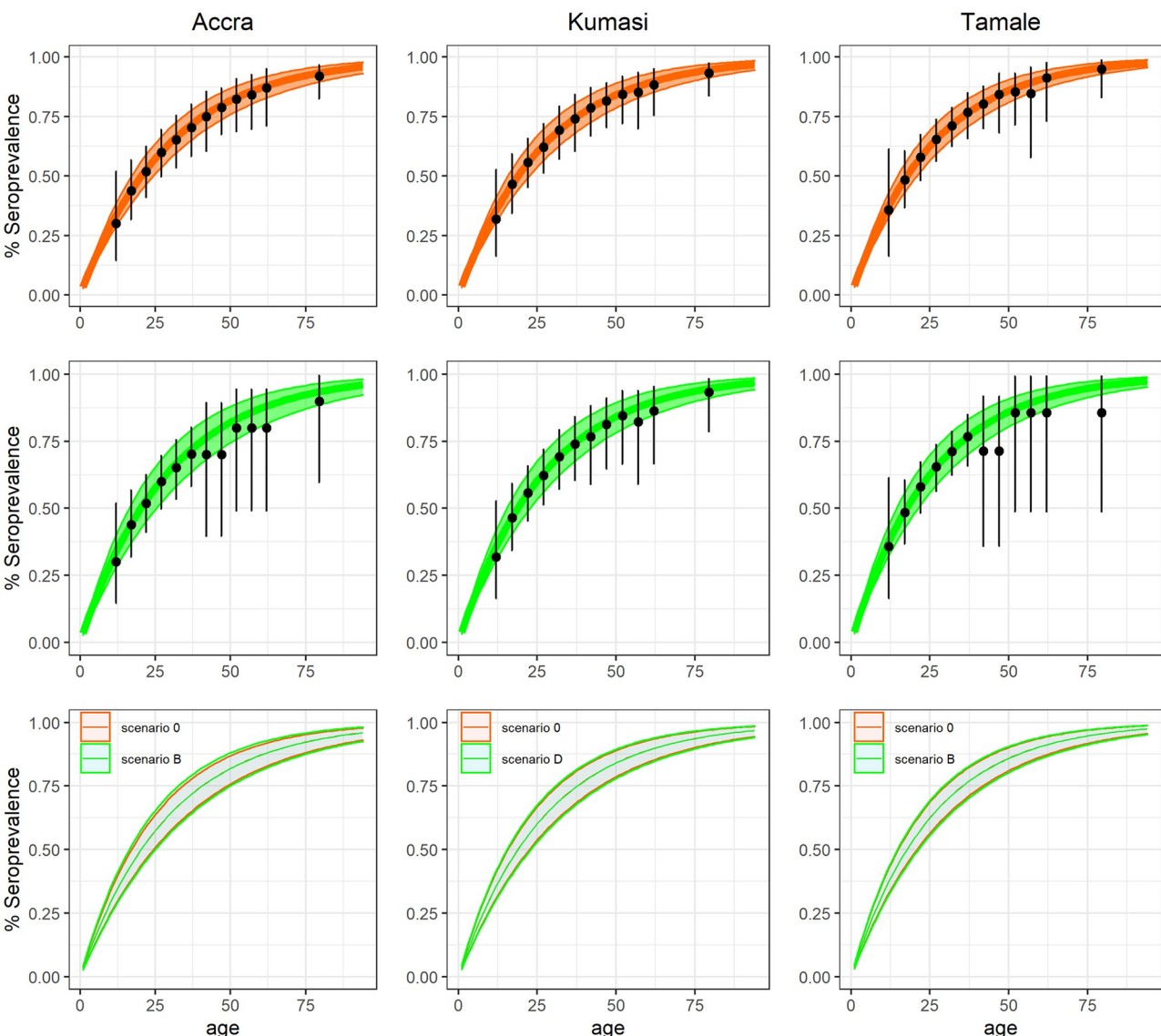

**Fig 3. Model fit for the three cities under the 5-year age categorization with model 1.** The first row represents the fit for scenario 0, where the median and 95% CrI of the estimated seroprevalence are reported in orange as a line and shaded area, respectively. The second row illustrates the model fit with the selected scenario in green. In both panels, the datasets used for model fitting are shown in black, with the error bars showing the exact binomial 95% Confidence Interval (CI). The third row shows a comparison of the estimated seroprevalence in scenario 0 (orange) and the optimal scenario (green).

precision and low uncertainty, as shown in Table 1. The model also fitted the simulated data, as shown in Fig F in S1 Text.

## The simulation framework works with decaying immunity and across transmission settings

The proposed simulation framework can successfully reconstruct the FOI also in scenarios with antibody decay and temporal changes in the intensity of transmission, including increasing, decreasing and non-monotonic transmission intensity patterns, and allows to identify an

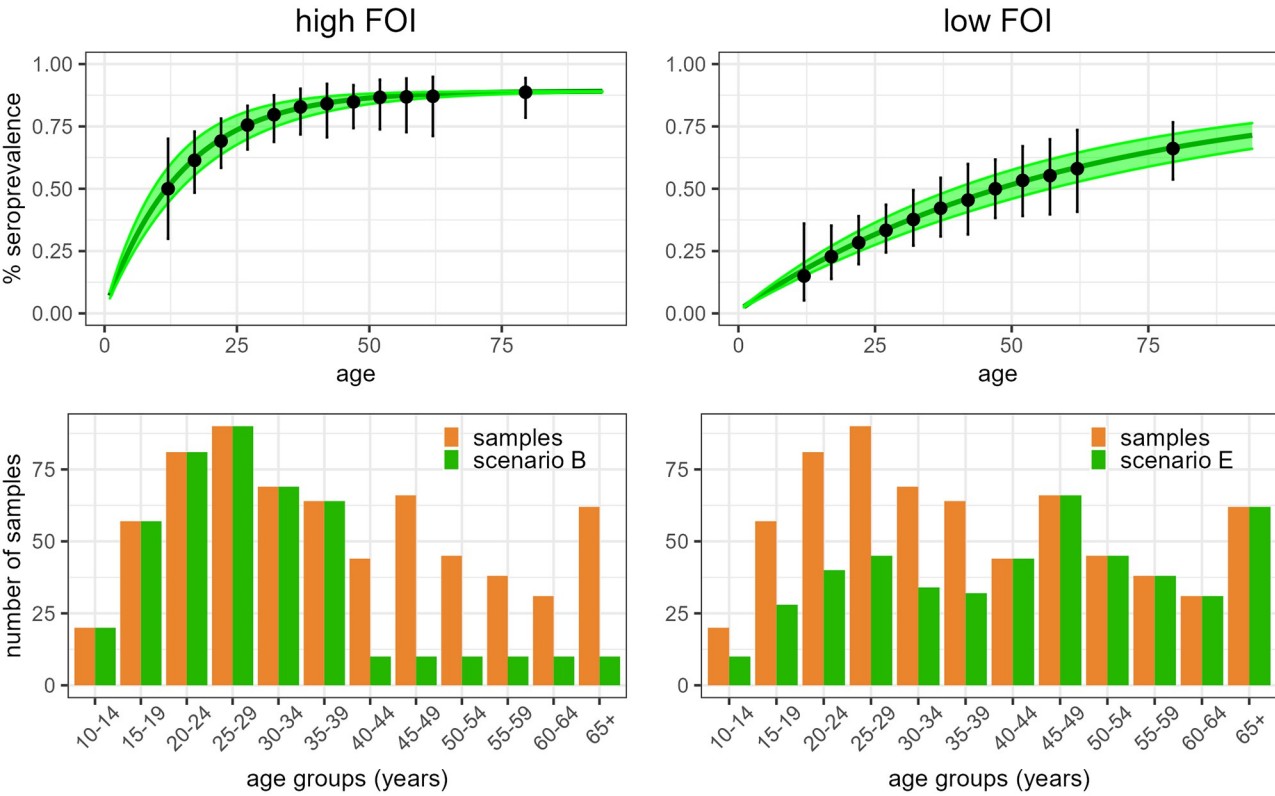

**Fig 4. Model fit for Accra under the 5-year age categorization and a high and low FOI.** The first row represents the seroprevalence using the chosen scenario, where the median and 95% credible interval of the estimated seroprevalence are reported in green as a line and shaded area, while the simulated dataset is reported in black, with the error bars indicating the exact binomial 95% CI. The second row illustrates the age-distributions of the tests with the chosen scenario (in green) for each city, compared to the available sample distribution in orange. The figure reports the results obtained with the high FOI (first column) and low FOI (second column).

optimal testing scenario also when using these alternative assumptions on immunity decay and temporal changes in the FOI.

The selected testing scenarios varied across locations and with the modelled assumptions (Table 1). The simulation-based framework under model 2 could capture the seroprevalence data (Fig G in S1 Text), and produced accurate estimates of the FOIs and decay rates (Fig D in S1 Text), with a slightly larger uncertainty around the decay rate than in the FOI. Scenario A, which includes the minimum number of tests performed across all age-groups, was among the preferred scenarios for the antibody decay model (model 2). However, when assuming a high FOI, the chosen scenario was B, which involves the testing of the younger age-groups.

We found preferential testing of older age-groups (scenarios C and E) when using the time-varying FOI model (model 3), which can be explained by the fact that older age-groups are the only ones providing information to estimate the earliest FOI ($\lambda_3$) in the time-varying model. The simulation framework can reconstruct the expected seroprevalence with good accuracy (Fig H in S1 Text) and provides precise estimates of all FOIs across the three time-varying scenarios explored, with a slight increase in uncertainty for the earliest FOI ($\lambda_3$) (Fig E in S1 Text).

## Discussion

Estimating the transmission intensity and burden of dengue globally is key to quantifying the current and future risk of infection and disease, and health care demand. In addition, it is

critical to assess the potential impact of new interventions, such as vaccination campaigns implemented alone or in combination with vector control interventions [26,27], both prospectively and retrospectively. Across Africa, only 17 seroprevalence surveys have been conducted to date [3,15] and, as such, the African region remains a continent with limited knowledge of the historical circulation and seroprevalence of dengue [8]. Serological surveys are, therefore, invaluable surveillance tools to help understand and estimate dengue transmission intensity and its heterogeneity across the continent [28].

We propose a simulation-based method to guide secondary testing of existing serosurveys for other pathogens. Retargeting existing serosurveys has been previously explored, for example by Carcelen et al. [29], who piggybacked on a HIV national serosurvey performed in Zambia to test for anti-measles and anti-rubella virus IgG antibodies, using almost 10,000 residual sera. While testing existing samples may not be suitable in all circumstances, there are several advantages to secondary testing of existing blood samples and the number of recent serosurveys conducted globally and across Africa during the COVID-19 pandemic provide new opportunities to assess arbovirus transmission [30,31]. Using existing samples from previous well-designed serosurveys reduces costs, resources and logistics, thus providing new opportunities for serosurveillance across multiple locations when the budget is limited. This approach is particularly important, given the limited resources available for public health initiatives in many parts of the world, and our study contributes to the development of new strategies to make best use of available resources and improve our understanding of viral seroprevalence.

The proposed framework can determine the testing scenario under different transmission intensity assumptions (constant FOI, time-varying FOI and with antibody decay) and estimate the FOI with high accuracy under the chosen scenario. Notably, different assumptions on the intensity of transmission as well as on the duration of immunity and on time-varying patterns of transmission influence the choice of the testing scenario. Moreover, we found that the FOI estimates obtained with model 2 (with antibody decay) and model 3 (with time-varying FOI) exhibit larger bias and uncertainty compared to the estimates obtained with model 1, which is expected given the larger number of parameters estimated using the same data.

In the context of the SERODEN survey in Ghana, we used the simplest constant FOI model due to the evidence that dengue antibodies are long lasting across several transmission settings [10]. Under this model, the testing scenarios selected in Ghana were scenario B for Accra and Tamale, and scenario D for Kumasi. Both scenarios require testing of the samples from young age-groups, which demonstrates the importance of recruiting children in seroprevalence studies. In the case of Kumasi, scenario B had a slightly higher bias and lower coverage compared to the baseline scenario, while scenario D not only decreased the bias but also increased the coverage while maintaining a low uncertainty around the estimate which made it the best one for this location. This demonstrates how the chosen testing scenario depends on both the transmission setting and the available number of samples.

One limitation of the simulation approach adopted in this study is the use of imputed FOI estimates from the first global FOI map for dengue developed by Cattarino et al. [3] which, despite the accurate calibration, was validated on a limited number of FOI estimates from the African region. In the absence of pre-existing information that can be used to inform the assumed FOI estimates, one possible strategy to overcome the intrinsic uncertainty associated with model estimates would be to implement serological surveys in two steps, e.g., by (1) testing half of the sample sizes identified in the simulation study according to the optimal age-distribution based on initial calculations, and these results can then be used to update the simulation study and inform the age-distribution, before (2) testing the other half of the samples (Fig 5). In S1 Text we provide an example of how it is possible to adjust the distribution of

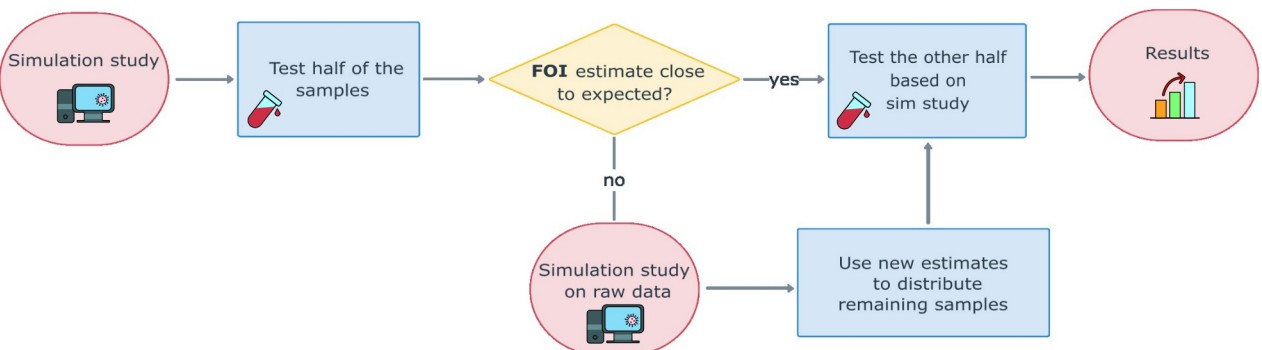

**Fig 5. Conceptual description of the workflow for the implementation of serological surveys in the absence of previous information (such as serosurvey or case-notification data) on the FOI.** Illustration of the two-step process proposed to identify the sample sizes and optimize their age-distribution when the study is conducted in locations with little or no prior information on dengue FOI.

the second half of the samples in a hypothetical situation where the FOI from the interim analysis (obtained from testing half of the samples in step 1) was different from the assumed one. We provided two examples exploring the case where the FOI estimate after step 1 was half the assumed one, and another where the FOI was twice the value assumed in the planning phase. The results are shown in the Supporting Information (Fig K in S1 Text), and in both cases, our framework and the two-step approach allowed us to efficiently adapt the testing scheme to the new information and re-distribute the remaining samples accordingly to accurately estimate the FOI (Figs K and L in S1 Text).

Another limitation of the method we developed is the empirical choice of the tolerance used to determine sample sizes such as the 15% difference from the upper bound of the 95% CrI of the baseline scenario which can be tweaked depending on the targeted accuracy and available budget and resources. Nonetheless, the methodological framework developed in this study provides a rationale that can be applied to secondary testing of existing blood samples, including in locations where there is limited previous information on the historical circulation and transmission intensity of a pathogen, while optimizing the accuracy of the FOI estimates and the use of resources.

## Conclusion

Age-stratified serological surveys are an ideal surveillance tool for reconstructing the age-dependent immunity profile of a population against one or more circulating viruses. Because the implementation of new serological surveys requires significant financial and human resources, testing stored blood samples (e.g., from previous serological surveys) can be an alternative strategy to estimate the FOI. In this work, we developed a method to inform the number of samples and age-distribution to test when using existing samples. In the presence of large uncertainties around the accuracy of the FOI estimates, we propose implementing the serological survey in two-steps, including an interim analysis of the results obtained upon testing half of the samples, which informs the ultimate sample distribution using the evidence collected from the transmission setting. The method developed in this paper can be used when testing existing blood samples for antibodies against different diseases with long-lasting or waning immunity as well as with time-varying transmission patterns, including in locations with limited prior information on the historical circulation of the pathogen of interest.

## Supporting information

**S1 Text. Includes Figs A-H, Tables A-I and additional details on the Two-steps sampling approach.**
(PDF)

**S1 Code. Incudes the code required to reproduce the analysis and results presented in this paper.** Within the "script" directory, the main analysis using Model 1 can be executed using sequentially numbered files 1 to 6. The analysis pertaining to the antibody decay (model 2) and time-varying FOI (model 3) can be run using the scripts numbered 7 to 10.
(ZIP)

## Author Contributions

**Conceptualization:** Belen Pedrique, Isabela Ribeiro, Gathsaurie Neelika Malavige, Christl A. Donnelly, Ilaria Dorigatti.

**Data curation:** Anna Vicco, Ilaria Dorigatti.

**Formal analysis:** Anna Vicco, Ilaria Dorigatti.

**Funding acquisition:** Belen Pedrique, Isabela Ribeiro, Gathsaurie Neelika Malavige, Ilaria Dorigatti.

**Investigation:** John H. Amuasi, Anthony Afum-Adjei Awuah, Christian Obirikorang, Nicole S. Struck, Eva Lorenz, Jürgen May.

**Methodology:** Anna Vicco, Christl A. Donnelly, Ilaria Dorigatti.

**Project administration:** Belen Pedrique, John H. Amuasi, Isabela Ribeiro, Gathsaurie Neelika Malavige, Ilaria Dorigatti.

**Resources:** Belen Pedrique, John H. Amuasi, Anthony Afum-Adjei Awuah, Christian Obirikorang, Nicole S. Struck, Eva Lorenz, Jürgen May, Isabela Ribeiro, Gathsaurie Neelika Malavige.

**Software:** Anna Vicco, Clare P. McCormack, Ilaria Dorigatti.

**Supervision:** Christl A. Donnelly, Ilaria Dorigatti.

**Validation:** Anna Vicco, Clare P. McCormack.

**Visualization:** Anna Vicco, Clare P. McCormack, Christl A. Donnelly, Ilaria Dorigatti.

**Writing – original draft:** Anna Vicco, Clare P. McCormack, Christl A. Donnelly, Ilaria Dorigatti.

**Writing – review & editing:** Anna Vicco, Clare P. McCormack, Belen Pedrique, John H. Amuasi, Anthony Afum-Adjei Awuah, Christian Obirikorang, Nicole S. Struck, Eva Lorenz, Jürgen May, Isabela Ribeiro, Gathsaurie Neelika Malavige, Christl A. Donnelly, Ilaria Dorigatti.

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
