## [Decision Letter · Decision Letter 0]

5 Jun 2023

Dear Dr Dorigatti,

Thank you very much for submitting your manuscript "A simulation-based method to inform serosurvey designs for estimating dengue force of infection using existing blood samples" for consideration at PLOS Computational Biology.

As with all papers reviewed by the journal, your manuscript was reviewed by members of the editorial board and by several independent reviewers. In light of the reviews (below this email), we would like to invite the resubmission of a significantly-revised version that takes into account the reviewers' comments.

Please note that this manuscript will likely need to be significantly updated / expanded to be accepted at PLOS Computational Biology. I am sympathetic to Reviewer 1's comments about the breadth of appeal of this work and its generality as a framework. If you prefer a more modest revision, submitting a revised version with reviews and responses to PLOS NTD may be a good alternative.

We cannot make any decision about publication until we have seen the revised manuscript and your response to the reviewers' comments. Your revised manuscript is also likely to be sent to reviewers for further evaluation.

Sincerely,

Alex Perkins

Academic Editor

PLOS Computational Biology

Thomas Leitner

Section Editor

PLOS Computational Biology

Reviewer's Responses to Questions

**Comments to the Authors:**

Reviewer #1: Vicco et al. describe a simulation approach for determining sample sizes and strategies when estimating the force of infection using serocatalytic models. The study focuses on estimating dengue FOI from age-stratified blood samples collected during the SARS-CoV-2 pandemic from 3 cities in Ghana; the crux of the problem being that the age-distribution of the samples was not optimised for dengue surveillance, and not all samples could be retested. The authors used a simulation-recovery approach to compare the bias, coverage and uncertainty of FOI estimates when using different subsets of the full sample set. The authors suggest that this approach could be used to optimize serosurvey design for other pathogen systems and propose a conceptual workflow for testing serum samples in two stages for more efficient test allocation.

The authors did a good job of explaining their sample selection method which will help researchers inform their serosurveys. The code is also available and very well documented, which is great. However, while technically sound, we feel that the study is very specific to their sample set and makes many simplifying assumptions. It might be worthwhile exploring how the optimal sampling design varies under different assumptions (waning immunity, varying force of infection, different mean age of infection, etc.) to demonstrate how this framework would work for different sample availabilities, epidemiology, FOIs etc. This would strengthen the author’s claim that this can be adapted to different pathogens and demonstrate how these findings are dengue-specific.

Major comments

1. Lack of literature context

The introduction is a great overview of dengue epidemiology and vaccine development. However, given that the paper is about simulation-guided serosurveys, there is a lack of literature review on studies that already exist in this space, and most of the information on dengue is not relevant. We are also sceptical of the claims of novelty here, as simulation-recovery is a standard tool for power calculations in seroepidemiology. The following studies address the task of optimizing serosurvey design for estimating epidemic dynamics (including simulations), arguably more thoroughly than the present study:

1. Vinh & Boni. Epidemics 2015 doi: 10.1016/j.epidem.2015.02.005

2. Sepúlveda et al. Malar J 2015 DOI: 10.1186/s12936-015-1050-3

3. Blaizot et al. BMC Medical Research Methodology 2019 https://doi.org/10.1186/s12874-019-0692-1

4. Larremore et al. eLife 2021 https://doi.org/10.7554/eLife.64206

5. The review article by Cutts & Handson https://doi.org/10.1111/tmi.12737 also has some good references.

Granted, Vicco et al. are investigating serocatalytic models, whereas most of these studies are focused on fitting compartmental models. I would suggest substantially reframing the introduction and discussion to place the study in its correct context, rather than focusing on dengue epidemiology which is arguably only incidental to this study.

2. Overly simple serocatalytic model

The serocatalytic model seems more suitable for a simple, single-variant pathogen rather than one as complex as dengue, given the epidemiology described in the introduction. For example, the authors refer to the literature on dengue serotype interactions (cross-reactivity in measured antibodies and temporary cross-protection between serotypes), but the model in equation 1 ignores these interactions and assumes that the force of infection is simply additive across serotypes (nλ). With this formulation, the n is irrelevant as it is just a fixed constant – we cannot disentangle the relative contribution of each serotype nor their cross-reactivity, and thus the estimated FOI is just an aggregate of all dengue activity (and there is complete cross-protection between serotypes). Other standard additions to the serocatalytic model are seroreversion and time-varying FOI, which could be explored in the simulation framework – it would be interesting to understand how the optimal sample set differs depending on the complexity of the fitted model. I did appreciate the inclusion of test sensitivity and specificity, though the robustness to different test accuracies could also be demonstrated. I would encourage the authors to explore these more complex models, both to demonstrate the application of this framework to different pathogens and to better capture the epidemiology of dengue.

3. Interpreting the differences between the 3 cities and lessons learned from different FOIs

The authors performed simulation-recovery experiments for the three cities; however, it does not seem like the parameter to be estimated (the FOI) differs much between them. Thus, we expected discussion as to why the chosen strategy differed for Kumsai and what lessons could be learned comparing the three cities. It would be more informative for readers to see how the optimal testing strategy differs for quite different FOIs and assumptions (e.g., very high and very low FOI; time-varying). For a constant FOI, it seems intuitive that preferencing testing in younger individuals would be optimal. There may be some neat results to be shown e.g., with age-of-first-infection. Does the optimal age-representation differ for a pathogen with a very young age-of-first-infection vs. later in life?

5. Figure 4 and iterative simulation-recovery framework

The iterative framework seems like a great idea, and it is interesting to think about how this 2-stage process might differ depending on our prior knowledge of the pathogen (e.g., entirely novel vs. new outbreak of well-understood pathogen). The authors could quite easily explore how this workflow would perform under different FOI assumptions using their simulation framework – simulate a representative serosurvey under various FOIs, and then perform the 2-stage uniform/tailored testing approach to find an optimal subset. This seems particularly pertinent given the authors rely on model-based FOI estimates as the ground truth for their dengue serosurvey – what if the true FOI is drastically different from the estimate from Cattarino et al, and the optimal sample distribution is different to the initial best-guess?

Minor comments

- Table 2 and Fig 1/Table 1 feel redundant together, suggest moving repeated information to the supplement and choosing one table or figure.

- It might be useful to briefly explain how the sampling frame of the original SARS-CoV-2 study was chosen.

- We assume that the age distribution from the World Population Prospect 2021 was used for the ground-truth simulation, but we did not see that mentioned in the methods.

- Given Fig1A, would it be informative to add an extra comparison scenario where the age-distribution of the samples matches that of the population? Would that provide a better ground-truth than scenario 0?

Reviewer #2: In this manuscript, Vicco and colleagues proposed a simulation-based method to assess the serosurvey designs using blood samples collected from previous surveys. Specifically, the authors applied their method to estimate dengue force of infection in Ghana, using samples previously collected for a SARS-CoV-2 serosurvey. The authors first simulated the age-dependent seroprevalence (Eq. 1) using the average yearly FOI per serotype that was estimated from Ref. [3], after which they used a binomial likelihood (Eq. 6) to reconstruct the posterior distribution of the force of infection. The authors tested several sampling scenarios to adjust the age distributions of samples.

The proposed statistical method looks reasonable, with a comprehensive simulation-based tests. I only have a few suggestions:

(1) Although using previous serosurvey provides a cost-effective data source for estimating dengue FOI, caution may be needed if two pathogens have some degree of cross-reactivity. For example, a few studies suggest the possible interaction between antibodies against SARS-CoV-2 and dengue virus [1]. The authors may briefly discuss measures to reduce this kind of bias:

Ref. [1]: Antibodies against the SARS-CoV-2 S1-RBD cross-react with dengue virus and hinder dengue pathogenesis, https://www.frontiersin.org/articles/10.3389/fimmu.2022.941923/full

(2) The authors assumed a uniform prior distribution between zero and one. Using weekly informative prior may be helpful to reduce the uncertainty.

(3) The authors may wish to briefly discuss the reason for choosing a tolerance of 15%.

**Have the authors made all data and (if applicable) computational code underlying the findings in their manuscript fully available?**

Reviewer #1: Yes

Reviewer #2: Yes

PLOS authors have the option to publish the peer review history of their article (what does this mean?). If published, this will include your full peer review and any attached files.

Reviewer #1: No

Reviewer #2: **Yes: **Lin Wang
---

## [Decision Letter · Decision Letter 1]

3 Oct 2023

Dear Dr Dorigatti,

Thank you very much for submitting your manuscript "A simulation-based method to inform serosurvey designs for estimating the force of infection using existing blood samples" for consideration at PLOS Computational Biology. As with all papers reviewed by the journal, your manuscript was reviewed by members of the editorial board and by several independent reviewers. The reviewers appreciated the attention to an important topic. Based on the reviews, we are likely to accept this manuscript for publication, providing that you modify the manuscript according to the review recommendations.

Please address the minor comments raised by the reviewer.

Sincerely,

Alex Perkins

Academic Editor

PLOS Computational Biology

Thomas Leitner

Section Editor

PLOS Computational Biology

Reviewer's Responses to Questions

**Comments to the Authors:**

Reviewer #1: The authors have done a very thorough job of addressing our comments, and the paper is now suitable for publication. Great job!

One last comment: I appreciate the addition of the two-stage process simulation. However, it is not currently clear how the two-stage process actually helps with estimating FOI. The point of our suggestion was to demonstrate how the two-stage process actually improves the FOI estimates -- the new additions simply shows how the sample distribution is updated. I would suggest adding a figure with FOI on the y-axis, and on the x-axis: 1) the true FOI; 2) the FOI assumed for the initial study; 3) the FOI estimated under the “Old_scenario” and 4) the FOI estimated under the “New_scenario”. I believe the authors are trying to show that the FOI estimate under the new scenario is more accurate than under the old scenario, but that doesn’t currently come across.

Minor points:

• L132: typo “reversible catalytic model” should be “models”

• L384, typo for “tests”

• Typo L389 "older age-groups are the only to"

• The bias values in Table 1 are order of magnitude 1-20, but the definition is “absolute value of difference between the estimate FOI and that used to simulate the data”. Surely the bias should be of order 0.01? Is what is being shown the percentage difference?

• It seems that with models 2 and 3, the bias and uncertainty both increase quite drastically – suggest discussing.

• Figure 3 is presumably for model 1 – please specify.

**Have the authors made all data and (if applicable) computational code underlying the findings in their manuscript fully available?**

Reviewer #1: Yes

PLOS authors have the option to publish the peer review history of their article (what does this mean?). If published, this will include your full peer review and any attached files.

Reviewer #1: **Yes: **James Hay

Figure Files:

Data Requirements:

Reproducibility:

References:

---

## [Editor Report · Decision Letter 2]

6 Nov 2023

Dear Dr Dorigatti,

We are pleased to inform you that your manuscript 'A simulation-based method to inform serosurvey designs for estimating the force of infection using existing blood samples' has been provisionally accepted for publication in PLOS Computational Biology.

Best regards,

Alex Perkins

Academic Editor

PLOS Computational Biology

Thomas Leitner

Section Editor

PLOS Computational Biology

---

## [Editor Report · Acceptance letter]

17 Nov 2023

PCOMPBIOL-D-23-00574R2 

A simulation-based method to inform serosurvey designs for estimating the force of infection using existing blood samples

Dear Dr Dorigatti,

I am pleased to inform you that your manuscript has been formally accepted for publication in PLOS Computational Biology. Your manuscript is now with our production department and you will be notified of the publication date in due course.

With kind regards,

Anita Estes
